# Role of Dendritic Cell in Diabetic Nephropathy

**DOI:** 10.3390/ijms22147554

**Published:** 2021-07-14

**Authors:** Hyunwoo Kim, Miyeon Kim, Hwa-Young Lee, Ho-Young Park, Hyunjhung Jhun, Soohyun Kim

**Affiliations:** 1Division of Nephrology, Department of Internal Medicine, College of Medicine, Jeju National University Hospital, Jeju National University, Jeju 63243, Korea; andrewmanson@naver.com (H.K.); spirit-rapper@hanmail.net (M.K.); doyul215@naver.com (H.-Y.L.); 2Research Group of Functional Food Materials, Korea Food Research Institute, Wanju 55365, Korea; hypark@kfri.re.kr; 3Technical Assistance Center, Korea Food Research Institute, Wanju 55365, Korea; 4Laboratory of Cytokine Immunology, Department of Biomedical Science and Technology, Konkuk University, Seoul 05029, Korea; 5College of Veterinary Medicine, Konkuk University, Seoul 05029, Korea

**Keywords:** chronic kidney disease, diabetic nephropathy, dendritic cell

## Abstract

Diabetic nephropathy (DN) is one of the most significant microvascular complications in diabetic patients. DN is the leading cause of end-stage renal disease, accounting for approximately 50% of incident cases. The current treatment options, such as optimal control of hyperglycemia and elevated blood pressure, are insufficient to prevent its progression. DN has been considered as a nonimmune, metabolic, or hemodynamic glomerular disease initiated by hyperglycemia. However, recent studies suggest that DN is an inflammatory disease, and immune cells related with innate and adaptive immunity, such as macrophage and T cells, might be involved in its development and progression. Although it has been revealed that kidney dendritic cells (DCs) accumulation in the renal tissue of human and animal models of DN require activated T cells in the kidney disease, little is known about the function of DCs in DN. In this review, we describe kidney DCs and their subsets, and the role in the pathogenesis of DN. We also suggest how to improve the kidney outcomes by modulating kidney DCs optimally in the patients with DN.

## 1. Introduction

Diabetic nephropathy (DN) refers to microvascular complications and end-stage renal disease in diabetic patients, accounting for approximately 50% of incident cases [1]. Clinically, DN is diagnosed based on the presence of albuminuria, decreased estimated glomerular filtration rate, or both, in patients with diabetes mellitus (DM) and histologically. DN is also characterized by glomerular hypertrophy, diffuse or nodular mesangial expansion, diffuse thickening of the glomerular basement membrane, glomerulosclerosis, podocytes loss, endothelial cell damage, tubular atrophy, interstitial fibrosis, arteriolar hyalinosis, and the infiltration of immune cells [2].

In general, DN has been considered as a nonimmune, metabolic, or hemodynamic glomerular disease initiated by hyperglycemia [2]. However, several other factors such as hypertension, hyperlipidemia, accumulation of advanced glycation end products (AGEs), and albuminuria contribute to the initiation and progression of DN [2,3]. In addition, growing evidence suggest that inflammatory elements and infiltrating immune cells in both the innate and adaptive immune systems play a significant role in the pathogenesis of DN [4,5,6,7,8,9,10,11]. Among immune cells, accumulation of macrophages is commonly observed in the glomeruli and kidney interstitium of both animals and humans with DN [12,13]. Infiltration of activated T cells is also described in the kidneys of patients with DM compared with those from patients with non-diabetic conditions, and the number of these cells is correlated with the degree of proteinuria in these patients [9,14], suggesting T cell mediated injury in DN. Dendritic cells (DCs) are antigen-presenting cells that link the innate and adaptive immune responses by inducing T-cell responses.

It has been also reported that high glucose leads to DCs activation and maturation [15,16], and the accumulation of these cells has been observed in animal models of DN and human kidney biopsies [7,17]. Moreover, it is known that kidney DCs are required to maintain the presence of activated T cells in the kidney disease [18]. However, little is known about the role of DCs in DN. In this review, we describe the subsets of kidney DC in the development and progression of DN, and suggest how to improve the kidney function by modulating kidney DCs immunologically in patients with DN.

## 2. Sources and Functions of Kidney Dendritic Cells

### 2.1. General Dendritic Cells

DCs are bone marrow-derived major antigen-presenting cells that bridge the innate and adaptive immune responses by inducing T-cell responses to captured exogenous or endogenous antigens, and several types of DCs have been identified: conventional (or classical) DCs (cDCs) which derive from myeloid precursors, plasmacytoid DCs (pDCs) originated from lymphoid precursors, and monocyte-derived DCs that differentiate from peripheral blood monocytes in response to inflammation [19,20].

DCs can be activated by the presence of pathogen-associated molecular patterns (PAMPs) such as lipopolysaccharide, or damage-associated molecular patterns (DAMPs) such as high-mobility group protein box 1 (HMGB-1). After sensing, capturing, and processing these molecules, DCs undergo maturation resulting in up-regulation of costimulatory molecules—such as CD80 and CD86—and chemokine receptors such as C-C chemokine receptor type 7 (CCR7), and secretion of various cytokines including tumor necrosis factor-alpha (TNF-α), interleukin (IL)-1 beta (IL-1β), IL-6, and IL-18. After this, they migrate to draining lymph nodes where they present processed antigen/peptide coupled to major histocompatibility complexes (MHCs), which allow for selection and expansion of antigen specific T cells [19,20].

In mice, cDCs express MHC class II and the integrin CD11c, and are further subdivided into cDC1 and cDC2 based on their expression of cell surface markers such as XC-chemokine receptor 1 (XCR1) and transcription factors such as basic leucine zipper transcription factor ATF-like 3 (Batf3) [21]. Murine cDC1s express several cell surface markers, the XCR1, CD103, and C-type lectin domain family 9 member A (CLEC9A; CD370) in non-lymphoid tissues and high levels of CD8 instead of the CD103 in lymphoid organs [19,22]. They are described to cross-present exogenous antigens on MHC class I molecules to CD8^+^ cytotoxic T cells (CTLs). In humans, cDC1s express CD141 (blood dendritic cell antigen 3; BDCA3) [23,24,25]. Murine cDC2s express the markers CD11b, CX_3_C-chemokine receptor 1 (CX_3_CR1; fractalkine receptor), and CLEC4A4 (dendritic cell inhibitory receptor 2; DCIR2).

In humans, cDC2s express CD1c (BDCA1) and some levels of CD1a [26,27]. Although cDC2s are known to be able to present antigen to both CD4^+^ and CD8^+^ T cells, they mainly process and present antigens on MHC class II to activate CD4^+^ T cells (T helper cells). In particular, they are inferior to cDC1s in cross-presentation of antigen derived from necrotic cells [24,28]. Mouse pDCs express high levels of plasmacytoid dendritic cell antigen-1 (PDCA-1), and human pDCs express the markers CD123 (interleukin-3 receptor), CD303 (BDCA2), and CD304 (BDCA4). Their unique function is sensing foreign or altered nucleic acids and the production of type 1 interferons during viral infections [29,30]. Monocyte-derived DCs are absent at the steady state and mainly arise in various inflammatory conditions [19,20].

### 2.2. Kidney Dendritic Cells-Types

Kidney DCs are a heterogeneous population composed of several subsets [31,32]. In normal human kidney tissue both cDCs and pDCs are in the renal interstitium and are rarely present within glomeruli, with the majority of DCs within the kidney being cDCs [32,33,34]. Murine kidney cDCs consist of CD103^+^ DCs (cDC1s) [35] which represents almost 50% of kidney DCs [36], and CD103^-^ DCs (cDC2s), which are characterized by expression of the CD11b and the chemokine receptor CX_3_CR1 [31]. In human, CD141^hi^ DCs (cDC1s) express lower HLA-DR and similar CD11c levels compared with CD1c^+^ DCs (cDC2s). CD1a is only present on a proportion of CD1c^+^ DCs, representing one of the migratory tissue DC populations [37].

Several growth factors, such as Fms-like tyrosine kinase 3 (Flt3) ligand and colony stimulating factor-1 (CSF-1), influence the development of DCs [35,38], and the migration into both healthy and inflamed organs, including the kidney, depends on chemokine receptors. It has been demonstrated that CCR1, CCR2, CCR5, CCR7, and CX_3_CR1 are expressed on kidney DCs. Among these chemokine receptors, their entrance into the healthy kidney is mediated by CCR5 [4]. Monocyte-derived DCs that differentiate from peripheral blood monocytes migrate into the injured kidney in response to inflammation via expressing the chemokine receptors CCR2 and CX_3_CR1 [39]. After activation, the migration of kidney DCs into the kidney drainage lymph nodes is dependent on CCR7 [4].

### 2.3. Kidney Dendritic Cells-Roles

In normal renal tissue, kidney DCs display an extensive and intricate parenchymal network around the tubule, interstitium, and glomeruli, and constantly survey their environment using their processes. When they encounter glomerular and tubular originated self-antigens and small molecular weight antigens that pass the glomerular filter from the tubular lumen, they remain immature and tolerize autoreactive T cells after migrating to the kidney lymph node, either by inducing T cell apoptosis or, in the case of T helper cells, by converting them into inducible T_reg_ cells (tolerization) [40,41]. That is, in the steady state, the expression of costimulatory molecules, such as CD80 and CD86, by kidney DCs is low, indicating they are immature and generally suppress adaptive immune responses by T and B lymphocytes [42].

During an acute infection, kidney DCs detect pathogens and contribute to the initiation of the host defense responses by attracting and activating innate immune cells, such as granulocytes or macrophages [43]. In this way, matured DCs activated by PAMPs, DAMPs, or inflammatory stimuli, produce cytokines and costimulatory molecules, and activate T cells and renal macrophages, thus mediating inflammation and kidney damage [44].

## 3. Pathologic Roles of Kidney Dendritic Cells in Diabetic Nephropathy

In chronic kidney disease (CKD), it has been demonstrated that DCs significantly increase in the interstitium of both experimental and human kidney diseases. Kidney DCs are important for the progression of these diseases including conditions that are not thought to be primarily induced by the immune system, such as hypertensive or obstructive nephropathy [36,37,45,46,47,48,49,50]. However, it is unlikely that all DC subsets cause kidney injury with the same pattern, since depletion of the whole DCs population in previous studies produced variable outcomes, sometimes opposing ones in different diseases [36,37,49,51]. Thus, it has been suggested that the roles of kidney DCs can differ according to the type and stage of the disease, and their subsets activated.

Previous reports with human kidney biopsy showed that there are significantly greater numbers of not only total DCs but also CD141^hi^ and CD1c^+^ cDC subsets (cDC1s and cDC2s, respectively) in diseased biopsies with interstitial fibrosis than diseased biopsies without fibrosis or healthy kidney tissue. However, pDC numbers are significantly higher in the fibrotic group compared with healthy tissue only. In addition, irrespective of primary kidney disease, there is an association between absolute numbers of tubulointerstitial CD141^hi^ CLEC9A^+^ DCs and CD1c^+^ DCs with both interstitial fibrosis and loss of kidney function. Moreover, transforming growth factor-beta (TGF-β) levels in dissociated tissue supernatants are significantly elevated in diseased biopsies with fibrosis compared with nonfibrotic biopsies, with cDCs identified as a major source of this profibrotic cytokine. This data suggested that activated cDC subsets, but not pDC, play a role in the development of fibrosis and, thus, progression to CKD [37].

In addition, studies in a murine model of human focal segmental glomerulosclerosis (FSGS) demonstrated that the number of kidneys CD103^+^ DCs (cDC1s) is significantly higher in mice with adriamycin nephropathy (AN) than in normal mice, and depletion of CD103^+^ DCs impairs activation and proliferation of CD8^+^ T cells and kidney injury. This study indicated that cDC1s, but not cDC2s or pDCs play a pathogenic role via the activation of CD8^+^ T cells in AN [36,49]. They also showed a significant increase in both CD141^+^ DCs (homologous to murine CD103^+^ DCs) and CD1c^+^ DCs in biopsies of patients with FSGS compared with healthy human kidney tissue. Moreover, CD141^+^ DCs increased more than CD1c^+^ DCs in patients with FSGS, indicating that cDC1s play a major role in progression of FSGS [49]. Taken together, all these data suggest that kidney cDC1s could have a pathogenic role in different kidney diseases such as FSGS.

Although relatively little is known about the role of DCs in the development and/or progression of DN, a few previous studies have suggested that DCs play a certain role during DN development and progression. Muller et al., in a nonimmune renal disease model using double-transgenic rats harboring both human renin and angiotensinogen genes, showed that angiotensin II leads to DCs accumulation in the kidney tissue and increases expression of MHC class II and CD86. This report implied that an activated renin-angiotensin-aldosterone system in the kidney tissue, frequently noted in patients with DM, is associated with kidney DC infiltration and maturation [45]. In addition, accumulation of CD1a^+^ CD80^+^ DCs in the renal tubules, interstitium and vessels, especially in interstitium, was also observed in a model of remnant kidneys, which is considered as non-immunologically mediated, and the accumulation extent of DCs was well associated with the loss of renal function and the progression of tubulointerstitial fibrosis [47].

An animal study using non-obese diabetic (NOD) mice demonstrated that CD4^+^ and CD8^+^ T-cells, and CD11c^+^ DCs infiltrate in the glomeruli, and the extent of infiltration of DCs correlates with albuminuria [7]. Another animal study using streptozotocin-induced diabetic rat model showed that compared with the control rats, expression of P-selectin (which is known as a cell adhesion molecule) is up-regulated in tubulointerstitium at week 4 in diabetic rats, followed by CD1a^+^ CD80^+^ DCs infiltration in renal tubulointerstitium and renal blood vessel, mainly in renal interstitium. They also found that the accumulation of DCs and expression of P-selectin is closely correlated with the degree of renal tubulointerstitial injury, suggesting that DCs play a role in tubulointerstitial damages in DN [52].

As for the roles of DC subtypes, previous studies have demonstrated that CD103^+^ DCs exhibit a renal pathogenic effect in murine CKD [36,49]. However, there is still limited information about the distinct functions of DC subsets in the pathogenesis of DN. Zhang et al. showed that the CD11c^+^ CD103^+^ DC subsets (cDC1s) significantly increase, express more costimulatory molecules such as CD80 and CD86 in DN rats compared with normal ones. CD11c^+^ CD103^+^ DC subsets enhance capacity of priming CD8^+^ T cell responses indicating that cDC1s might play an important role in the progression of renal injury in DN [53]. Furthermore, in DN rat mesenchymal stem cells (MSCs) transplantation significantly decreased CD103^+^ cDC1s, but not CD11b+ cDC2s, reduced proinflammatory cytokines such as IL-1β, IL-6, TNF-α and monocyte chemoattractant protein-1 (MCP-1), which diminished renal injury and fibrosis through reduced infiltration in the kidney and inactivation of renal CD8^+^ T cells. These results suggest that CD103^+^ cDC1s, but not cDC2s, might play an important role in the progression of renal injury in DN, and they might be the potential target for DN treatment as shown in Figure 1. In addition, the roles of kidney dendritic cells in diabetic nephropathy are summarized in Table 1.

## 4. Mechanisms of Kidney Dendritic Cells Activation in Diabetic Nephropathy

There are several plausible mechanisms by which diabetic environment may activate kidney DCs and subsequently mediate tubulointerstitial injury in DN. First, hyperglycemia per se is a likely candidate. Studies in cell cultures stimulated with high glucose demonstrated that the expression of CD83 and CD86 in DCs is up-regulated, and the production of proinflammatory cytokines, such as IL-6 and IL-12, increases via several cell signal transduction pathways such as protein kinase C [3] and p38 MAPK [16] in monocyte derived human DCs, leading to their maturation and activation [15,16]. Second, in diabetes AGEs and modified proteins excessively accumulate and bind to AGE-specific receptors on kidney DCs and subsequently activate them [54,55,56]. Third, the intra-renal renin-angiotensin-aldosterone system activation, especially angiotensin II induced by diabetic environment, may lead to kidney DCs activation and maturation in DN [45,57,58]. Fourth, the renal expression of the main proinflammatory cytokines—including TNF-α, IL-1β, IL-6, and IL-18—produced and secreted by immune cells as well as resident renal cells is increased in DN [6,59,60], which subsequently can trigger kidney DCs differentiation and maturation. Fifth, in proteinuric state, filtered high molecular weight proteins such as albumin can be transferred to DCs by renal tubular cells [61]. After this, these proteins can be a source of potentially antigenic peptides and kidney DCs, especially cDC1s, which come to play a role in processing these protein fragments into smaller peptides through a proteasome-dependent pathway. Consequently, immunogenic DCs can cross-present these peptides on MHC class I, to CD8^+^ T cells and trigger their effector response [18,51,61]. That is, through the proteasome, kidney DCs process most high molecular weight protein fragments into smaller peptides, which are presented to CD8^+^ T cells, leading to their activation.

Finally, several endogenous DAMPs, including Tamm Horsfall glycoprotein, heat shock protein 70, HMGB-1, and hyaluronan fragments that are generated during DM, can promote inflammatory responses by binding to toll-like receptor (TLR)2 and/or TLR4 expressed on DCs [36,62,63,64]. The best studied example is HMGB-1, which is a nuclear protein that is released into extracellular fluid in response to cell stress and injury, and it has been reported that HMGB-1 can directly activate CD103^+^ DCs in AN mouse [36]. Furthermore, increasing evidence indicates that HMGB-1 has also an important pathological role in DN by activating TLR2 and TLR4 signaling [65,66]. In summary, under the diabetic condition, DAMPs released by damaged tissues and extracellular matrix components bound to and activate pattern recognition receptors on kidney DCs, and subsequently results in the synthesis and secretion of proinflammatory cytokines and some chemokines, culminating in activation of CTLs via DCs.

Why is it that cDC1s, rather than cDC2s or pDCs, are associated with the pathogenesis of DN? DM associated stimuli that specifically activate cDC1s are still unknown and it remains to be clarified. However, it is known that compared with other DC subtypes, cDC1s can very efficiently phagocytose via endocytic receptors such as CLEC9A, process and cross-present cell-bound or soluble antigens from injured cells on MHC class I to CD8^+^ T cells [67,68]. Thus, the role of cDC1s in the progression of DN may be due to their superior, specialized cross-presentation ability to exclusively internalize antigens derived from diabetic environment [69,70].

## 5. Interventions of Dendritic Cell Function in Diabetic Nephropathy

Although the regulation mechanisms of activation and infiltration of DCs in renal tissue during initiation and progression of DN is still unclear, several treatment options that can interfere with the functions of kidney DCs, especially cDC1s, could be useful in DN.

### 5.1. Mesenchymal Stem Cells Transplantation

MSCs have been widely used to treat animal model of DN in the last decade because multiple studies showed that MSCs reduce the expression of MHC class II, CD40, and CD86 costimulatory molecules on mature DCs, which is responsible for a decrease in T cell proliferation [71,72,73,74,75] and inhibit kidney damage through decreasing inflammatory conditions induced by high glucose [76,77]. However, the potential protective mechanisms for MSC-based therapy associated with DCs in DN remain obscure.

A recent animal study suggested a plausible mechanism as follows: bone marrow derived MSCs transplantation considerably decreased pro-inflammatory cytokines including IL-1β, IL-6, TNF-α, and MCP-1, subsequently diminished renal injury, and fibrosis, and recovered kidney function in DN rats. In addition, the MSCs-treated DN rats had decreased number and activity of CD103^+^ DCs and CD8^+^ T cells, but not CD11b^+^ DCs in the kidney. Moreover, TLR2 and TLR4 expression on CD103^+^ DCs was reduced when these cells were cultured with MSC conditioned media. Thus, these results suggest that the protective effect of MSCs may be related to their immunosuppression of CD8^+^ T cell proliferation and activation selectively mediated by CD103^+^ DCs in the kidney of DN rats [53]. Although the exact nature of the immune response to MSCs still remains poorly characterized, this study indicates a potential mechanism of renal protection that regulating CD103^+^ DCs can be a promising therapeutic strategy to prevent or treat DN.

### 5.2. Fms-Like Tyrosine Kinase 3 Ligand Inhibition

Flt3 is a receptor that is expressed with high specificity on tissue resident CD103^+^ DCs (cDC1s) and Flt3 ligand is a growth factor essential for the development and survival of CD103^+^ DCs. Therefore, CD103^+^ DCs are critically dependent on Flt3, and the development of CD103^+^ DCs is more strongly inhibited in Flt3- and Flt3L-deficient mice than other subsets of DCs [35,78], suggesting Flt3 inhibition can be an option to suppress cDC1s activity in the kidney disease. As in the aforementioned FSGS animal study, Flt3 inhibitors were shown to selectively deplete CD103^+^ DCs (but not CD103^-^ DCs or pDCs) and subsequently, alleviated renal injury through suppression of CD103^+^ DC-mediated CD8^+^ T cell activation in AN mouse [36]. Furthermore, a recent study showed that selective Flt3 inhibitor, AC220, effectively suppresses CD103^+^ DCs, but not CD103^-^ DCs or pDCs and reduces kidney injury in AN mouse, indicating that this inhibitor more strongly suppresses the development of CD103^+^ DCs than other subsets. AC220 might be a useful pharmaceutical agent to treat AN, in which CD103^+^ DCs have a critical role in the progression [49]. In rat DN, CD103^+^ DCs also showed increased mRNA expression of Flt3, which was reduced by MSCs transplantation and was associated with improved kidney function. Thus, these data suggest that similar with the AN model, Flt3 inhibition may be an option to prevent the progression of DN [53].

### 5.3. Proinflammatory Cytokine Inhibition

It is known that the renal expression of the main proinflammatory cytokines—such as IL-1β, IL-6, and IL-18, as well as TNF-α—by glomerular resident and interstitial cells in the renal tissue of patients with DN are increased [6,59,60], and kidney DCs can be activated by these cytokines released from these cells. Thus, blocking these proinflammatory cytokines can be a method to attenuate kidney DCs activation. Among the DN-associated proinflammatory cytokines anti-IL-1β and anti-TNF-α, have been shown to reduce albuminuria and kidney injury in db/db mice or streptozotocin mouse model, and in streptozotocin-induced diabetic rats, respectively [11,79]. In addition, in a small case series, inhibition of IL-1β improves kidney function transiently in patients with gout, DM, and moderate-to-severe CKD [80]. However, the effect of anti-TNF-α and anti-IL-1β on kidney DCs were not directly examined. Therefore, work for their efficacy in DN, especially the effects on kidney DCs, is needed in the future.

### 5.4. Toll-Like Receptors Ligand’s Inhibition

Diabetic environment may produce several endogenous DAMPs [36,62,63,64]. Among them, it is known that HMGB-1 can directly activate CD103^+^ DCs via HMGB-1 receptor or TLR4 expressed on these cells, and blockade of HMGB-1 (CD103-saporin antibody) results in a decrease in expression of CD80 and CD86 in CD103^+^ DCs in AN mouse [36]. Thus, it is likely that the inhibition of CD103^+^ DCs with HMGB-1 inhibitor may be a potential therapeutic approach to attenuate kidney inflammation and injury in CD103^+^ DCs mediated CKD, such as AN and DN. In line with this suggestion, recent animal studies demonstrated that blockade of HMGB-1 attenuates DN in streptozotocin induced diabetic rat and mice models, showing reducing albuminuria, glomerular injuries, interstitial fibrosis, and renal inflammation, although the association between HMBG-1 inhibition and functional changes in kidney DCs was not evaluated [81,82].

### 5.5. Proteasome Inhibition

Podocyte or renal mesangium derived antigens may be shed, reabsorbed from the glomerular filtrate, and acquired, processed, and presented by tubulointerstitial kidney DCs to CD8^+^ T cells. As we discussed above, in DCs, processing these protein fragments into smaller peptides are conducted by the proteasome pathway [83], which is activated in DN [84]. Thus, proteasome inhibition in kidney DCs may be a therapeutic target for DN. One experimental study using bone marrow derived DCs from rats and renal mass ablation proteinuric progressive nephropathy model showed that the proteasome inhibitor lactacystin or bortezomib completely prevents the stimulation of CD8^+^ T cells by DCs pre-exposed to albumin fragment [61]. In addition, more specific proteasome inhibitors such as MG132 also showed cytoprotective effect in DN, although these studies were conducted in human mesangial cell lines, but not DCs [85,86]. Thus, direct evidence of DC-targeted agents in DN is still lacking.

### 5.6. Scavenger Receptors Inhibition

Dioscin is a natural steroidal saponin that is isolated from certain medicinal plants or vegetables, and exhibits cytoprotection by reducing reactive oxygen species production, inflammatory cytokines secretion, and oxidized low-density lipoprotein uptake by DCs through preventing the expression of scavenger receptors such as CD204, CD36, and lectin-like oxidized low-density lipoprotein receptor-1, and p38 mitogen-activated protein kinase under high glucose conditions [16]. Thus, these results suggest that dioscin may be a therapeutic option to reduce cDC1s, which are known to have high levels of endocytic receptors such as CLEC9A [24].

In conclusion, the current treatments are inadequate due to the lack of understanding for the precise mechanisms of DN development and progression. Although, optimal control of hyperglycemia and elevated blood pressure with anti-diabetic drugs and renin-angiotensin-aldosterone system blocking agents (such as angiotensin-converting enzyme inhibitors, angiotensin II receptor blockers, or aldosterone receptor blockers) has been demonstrated to be effective in slowing the progression of DN. Among the several risk factors for DN, there is a growing body of evidence that kidney DCs, especially cDC1s, participate in the renal injury under the conditions of DN, and their activation in the kidney may be a crucial step in the progression of this disease. The knowledge of DCs’ role in kidney injury under DM could lead to the development of new and novel DC-based therapies for the prevention and treatment of DN in the future.

## Figures and Tables

**Figure 1 ijms-22-07554-f001:**
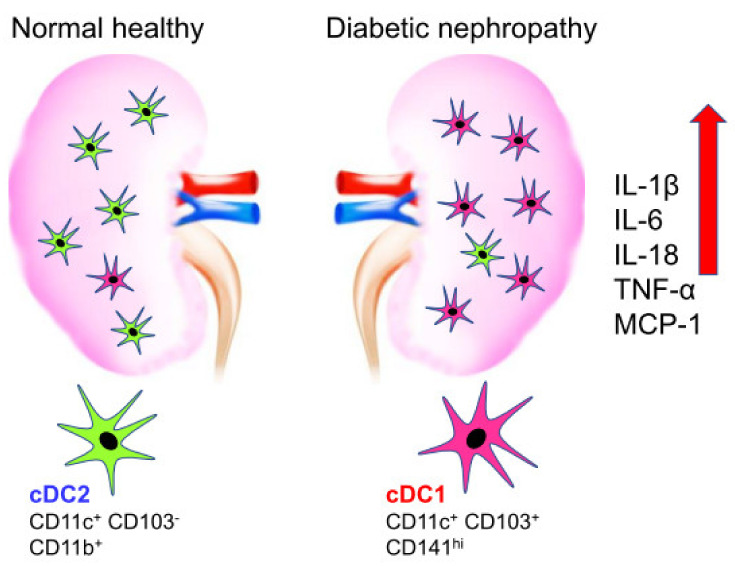
Difference in the population of dendritic cell in normal healthy compared to diabetic nephropathy. Normal healthy kidney has CD11c^+^ CD103^−^ DC2s on the left whereas diabetic nephropathy increased CD11c^+^ CD103^+^ CD141^hi^ DC1s on the right. The induction of inflammatory cytokines is observed in diabetic nephropathy on the right.

**Table 1 ijms-22-07554-t001:** Roles of kidney dendritic cells in diabetic nephropathy.

Models	Results	Reference
**Non-obese diabetic mice**	CD11c^+^ DCs have close contact with CD4^+^ and CD8^+^ T cells in the glomeruli.	[7]
**NOH transgenic mice**	DCs in renal lymph nodes constitutively cross-present ovalbumin and activate CD8^+^ T cells	[18]
**Human cord blood derived DCs**	Angiotensin II promotes the maturation and infiltration in the renal interstitial of DCs via TNF-α leading to albuminuria and renal fibrosis.	[45]
**Rat remnant kidneys induced by subtotal nephrectomy**	The accumulation extent of DCs is associated with the loss of renal function and the progression of tubulointerstitial fibrosis.	[47]
**Streptozotocin-induced diabetic rat (1)**	CD1a^+^ CD80^+^ DCs accumulation in the renal tissue and the extent of DCs accumulation are closely correlated with the degree of renal tubulointerstitial injury.	[52]
**Streptozotocin-induced diabetic rat (2)**	CD11c^+^ CD103^+^ DC subsets increase, express more costimulatory molecules, and enhance capacity of priming CD8^+^ T cell responses	[53]
**Rat proximal tubular cells**	Albumin is a source of antigenic peptides and trigger CD8^+^ T cells after being transferred to and processed by DCs through a proteasome-dependent pathway.	[61]

## Data Availability

This study did not report any data.

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
