# Peer review of "Role of Dendritic Cell in Diabetic Nephropathy"

_ijms, 2021, doi:10.3390/ijms22147554_

Round 1
Reviewer 1 Report
This review is very timely, as diabetic nephropathy (DN) is increasing world-wide due to the rising burden of Type 2 diabetes (T2D). Other recent reviews, while mentioning the role of dendritic cells, have mainly covered macrophages, T cells, B cells, mast cells, and neutrophils. Dendritic cells are of particular interest here, as they link innate and adaptive immunity by inducing T-cell responses. The authors have focused on describing specific dendritic cell subsets found in the kidney, and pinpoint which are most likely to contribute to the pathogenesis of DN. They also offer and discuss suggestions for future research, and for intervening at different aspects/steps of kidney DC functions to mitigate the development and/or progression of DN.
There are some clarifications and corrections needed:
1) Please clarify (line 227) that it is hyaluronan fragments that are generated during DM which can then promote inflammatory responses etc.
2) line 334: should be "CLEC9A" not CLECA9.
Author Response
Dear Reviewer,
We corrected them according to your suggestion. The changes were highlighted by yellow color.
1) Please clarify (line 227) that it is hyaluronan fragments that are generated during DM which can then promote inflammatory responses etc.
Answer: Corrected it.
2) line 334: should be "CLEC9A" not CLECA9.
Answer: Corrected it.
Reviewer 2 Report
They well summarized about dendritic cells (DCs) from their basic information to their possible functions in diabetic nephropathy (DN). Although the present manuscript is enough informative, a table to summarize the possible functions of DCs in DN and a figure to illustrate the mechanism to activate DCs would be very helpful for readers.
Author Response
Dear reviewer,
We also added new figure and table according to your suggestion. The changes were highlighted by yellow color.
They well summarized about dendritic cells (DCs) from their basic information to their possible functions in diabetic nephropathy (DN). Although the present manuscript is enough informative, a table to summarize the possible functions of DCs in DN and a figure to illustrate the mechanism to activate DCs would be very helpful for readers.
Answer: We added figure and table at the end of section three (3. Pathologic roles of kidney dendritic cells in diabetic nephropathy) in page 14.